# Durability of Implanted Low-Density Polyacrylamide Hydrogel Used as a Scaffold for Microencapsulated Molecular Probes inside Small Fish

**DOI:** 10.3390/polym14193956

**Published:** 2022-09-22

**Authors:** Ekaterina Shchapova, Evgeniy Titov, Anton Gurkov, Anna Nazarova, Ekaterina Borvinskaya, Maxim Timofeyev

**Affiliations:** 1Institute of Biology, Irkutsk State University, 664025 Irkutsk, Russia; 2Baikal Research Centre, 664003 Irkutsk, Russia; 3East Siberian Institute of Medical and Ecological Research, 665827 Angarsk, Russia

**Keywords:** polyacrylamide, hydrogel, non-biodegradable, implant, fish, *Danio rerio*

## Abstract

Implantable sensors based on shaped biocompatible hydrogels are now being extensively developed for various physiological tasks, but they are usually difficult to implant into small animals. In this study, we tested the long-term in vivo functionality of pH-sensitive implants based on amorphous 2.7% polyacrylamide hydrogel with the microencapsulated fluorescent probe SNARF-1. The sensor was easy to manufacture and introduce into the tissues of a small fish *Danio rerio*, which is the common model object in biomedical research. Histological examination revealed partial degradation of the gel by the 7th day after injection, but it was not the case on the 1st day. Using the hydrogel sensor, we were able to trace the interstitial pH in the fish muscles under normal and hypercapnic conditions for at least two days after the implantation. Thus, despite later immune response, amorphous polyacrylamide is fully suitable for preparing implantable sensors for various mid-term physiological experiments on small fishes. The proposed approach can be further developed to create implantable sensors for animals with similar anatomy.

## 1. Introduction

Multiple fields of modern science and technology demand new tools for convenient real-time monitoring of various physiological parameters inside animal tissues [1,2]. Most of the critical parameters are difficult to fully track non-invasively [1,3], and implantable miniaturized sensing tools have drawn more and more attention for these cases [4,5,6]. Various completely implantable sensors may provide a promising compromise between the need for repetitive measurements and minimizing traumas during skin punctures, along with the risk of associated infection. It is especially relevant for such animals as pelagic fishes, because their active movement interferes with long-term installation of wired probes, while every blood sampling induces significant stress and requires anesthesia. At the same time, many species of fish are of interest for the growing market of aquaculture [7] and used as model objects in various fundamental and environmental research [8,9].

Despite a multitude of electronics-based implantable sensors being available [4], electronics-free alternatives have also been extensively developed due to their generally simpler preparation protocols [10]. Usually, such alternatives transmit information via light and consist of a fluorescent sensitive component with high selectivity for the parameter of interest and some polymeric carrier [6]. In particular, the carriers are required to isolate the chemical probes from the animal tissues and thus to ensure their prolonged functionality. The choice of the carrier is non-trivial and depends on the animal anatomy and properties of the specific organ. For example, layer-by-layer assembled polyelectrolyte microcapsules (PMs) with soft walls are known to distribute well in the bloodstream [11] and, thus, can be used for physiological measurements inside blood capillaries [12]. At the same time, PMs are sub-optimal for application in animal tissues. Even though a slurry of PMs carrying a molecular probe can be easily injected into fish muscles and used for in vivo measurements within a few hours [13], relatively soon they are recognized as foreign bodies by immune cells and engulfed by phagocytes [12], which leads to incorrect readings.

In the case of subcutaneous or intramuscular applications inside relatively large animals, hydrogels are a common choice as the material for implantable optical sensors [6,14,15]. Various polymers can be used to fabricate such hydrogel-sensing implants. Natural polymers such as collagen, alginate, fibrin, hyaluronic acid, and chitosan and dextran in their natural form are used in many in vivo applications, but their disadvantage as an implant material is rapid degradation by enzymes [16,17]. Therefore, synthetic polymers are the most common choice to make the implants long-lasting [6,18]. Polyethylene glycol, poly-hydroxyethyl methacrylate, and acrylic acid are among the most commonly used synthetic biocompatible polymers [19,20,21,22], which in their conventional formulations form shaped hydrogels after polymerization.

However, we have encountered difficulties in implanting shaped hydrogels into small fish. In particular, surgical implantation would be an unnecessarily complicated procedure in this case. Alternatively, injection of the dense gel is difficult, as with the shallow injection depth the tight skin and muscles of the fish tend to push the thick gel back, while making a sufficiently thin gel is technically challenging. Another option for small animals is to reduce the density of the hydrogel to make it amorphous and easy to inject with a small needle. At the same time, a lower concentration of the polymer chains may make the hydrogel more structurally vulnerable to the immune cells, which potentially may disintegrate or even penetrate the implant and influence the readout of the molecular probe if it is dispersed directly in the gel. Thus, in this case it may be favorable to firstly anchor the fluorescent probe inside PMs or similar microcarriers and then embed these PMs into the semi-liquid hydrogel in order to give the probe double protection. This approach of combining microcapsules with hydrogel scaffolds has been previously applied for different biomedical tasks [23,24,25] but not for sensing purposes. The amorphous but viscous hydrogel carrier in tissues should remain integrated as one piece better than free PMs and is expected to delay their immune recognition. Importantly, for commonly used implantable synthetic polymers, it is possible to tune their physical and chemical properties by adding and substituting copolymers, but this implies complex chemical synthesis that is difficult to perform in most biological laboratories [20,21,26,27,28].

In this study, we tested the immunocompatibility and durability of amorphous low-density polyacrylamide inside the muscles of zebrafish *Danio rerio* (Hamilton, 1822), which is a common model object in numerous biomedical works [8]. Polyacrylamide was chosen due to its easy preparation protocol and wide availability in biochemical laboratories worldwide. In particular, we used polyacrylamide hydrogels carrying polyelectrolyte microcapsules (PAAH + PMs) containing the pH-sensitive dye SNARF-1 (conjugated with dextran). We analyzed short-term and long-term reactions to this pH sensor in fish tissues using histological analysis and further tested its responsiveness to changes of interstitial pH.

## 2. Materials and Methods

### 2.1. Materials

All reagents used for the preparation of polyacrylamide hydrogels with pH-sensitive fluorescent microcapsules and further procedures were of analytical purity and were used without additional purification. Acrylamide (#A1089) was purchased from AppliChem GbmH (Darmstadt, Germany). The following reagents were purchased from Helicon (Moscow, Russia): N,N′-methylenebisacrylamide (#22797959), ammonium persulfate (APS, #H-248614), N,N,N′,N′-tetramethylethane-1,2-diamin (TEMED, #68604730). The conjugate of seminaphtharhodafluor-1 with dextran (SNARF-1-dextran; #D-3304) was bought from Thermo Fisher Scientific (Eugene, OR, USA). Poly (allylamine hydrochloride) (PAH; #283215) and poly (sodium 4-styrenesulfonate) (PSS; #243051) were provided by Sigma-Aldrich (produced in St. Loius, MO, USA and Overijse, Belgium, respectively).

### 2.2. Preparation of pH-Sensitive Hydrogel

Encapsulation of the fluorescent dye was performed by layer-by-layer adsorption of oppositely charged polyelectrolytes, as described in detail previously [29,30]. Calcium carbonate seeds containing the fluorescent dye were prepared by mixing 0.625 mL of 1 M NaCO_3_ solution, 0.625 mL of 1 M CaCl_2_ solution, and 2 mL of 2.5 mg/mL pH-sensitive SNARF-1-dextran with constant stirring at room temperature. The resulting microparticles were sequentially immersed in 4 mg/mL solution of positively charged PAH (containing 1 M NaCl), washed with normal saline, immersed in 4 mg/mL negatively charged PSS (with 1 M NaCl), and washed again with saline. After placing 12 layers of the polymers, the carbonate cores were dissolved in 0.1 M EDTA solution at pH 7.0, resulting in hollow polyelectrolyte microcapsules (PMs) with a wall formula (PAH/PSS)_6_ loaded with the fluorescent dye.

A pH-sensitive 2.7% polyacrylamide hydrogel (PAAH) was prepared in a laminar flow cabinet to maintain sterility (Figure 1A). For this, in a typical procedure, 300 μL of a suspension of microcapsules were mixed with SNARF-1-dextran in saline (7 × 10^5^ microcapsules per 1 μL; water without PMs was added for further testing of PAAH physical properties), 30 μL of 30% acrylamide solution with 0.8% bis-acrylamide, 3 μL of 10% APS, and 3 μL of TEMED in a 0.5 mL microtube. The mixture was drawn into an insulin syringe, and air bubbles were dispelled to allow polymerization. Polyacrylamide hydrogel with fluorescent microcapsules (PAAH + PMs) was polymerized for 30–40 min at 21 °C with constant rotation of the syringe on an Intelli-Mixer RM-1L orbital mixer (ELMI, Riga, Latvia) to prevent the microcapsules from settling.

Then, 1 M Na_2_HPO_4_ buffer solution (pH 4.5, passed through a 0.2 μm filter to sterilize) was drawn into a syringe with hydrogel (1/3 of the gel volume) and incubated for 15 min. The procedure was repeated 3–4 times until the pH of PAAH was adjusted to the physiological level of 7.4–7.6 (pH measurements are described below). Finally, polyacrylamide hydrogel with pH-sensitive polymeric microcapsules was additionally washed from unpolymerized components by drawing 0.9% sterile saline into the syringe at least 10 times and was stored at +4 °C until use.

A kinematic viscosity of 2.7% PAAH without PMs was roughly estimated by the gravimetric method using glycerol with purity over 98% (Vekton, Saint Petersburg, Russia) as the reference. Eight milliliters of both substances were allowed to freely flow inside the same vertical plastic tube with the inner diameter of 13 mm at a temperature of 21 °C, and the process was monitored by a Tough TG-5 camera (Olympus, Tokyo, Japan) in high-speed mode in order to increase the precision of the time measurements. According to the available literature, at room temperature glycerol has a kinematic viscosity of approximately 1160 mm^2^/s [31], and the value for 2.7% PAAH was calculated proportionally to the ratio of times required for two substances to pass the same distance.

### 2.3. Animal Maintenance

All experimental procedures were conducted in accordance with the EU Directive 2010/63/EU for animal experiments and the Declaration of Helsinki; the protocol of the study was registered and approved before the start of the experiment by the Animal Subjects Research Committee of the Institute of Biology at Irkutsk State University (Protocol No. 2017/1). The study used a depigmented adult *D. rerio* with a body length of 2.5–3 cm, purchased from a local pet store. The fish were kept in aerated aquaria with the volume of 60 L (50 animals per aquarium) in tap water filtered through an AquaPro filter system (mechanical cleaning cartridge EFG-63/250, pressed carbon ARS, granulated carbon UPF) at room temperature. Animals with implanted pH-sensitive hydrogels were kept individually in 5 L aquaria with aeration. The fish were fed commercial feed, TetraMin Flakes (Tetra, Melle, Germany), every day at 0.02 g per fish.

### 2.4. Injection of Polyacrylamide Hydrogel

Before the implantation procedure, the fish were placed in an aqueous suspension of clove oil (0.05 mL/L) until the animal was completely immobilized (2–6 min). The individuals were then fished out from the aquarium with a net and placed on a glass slide. An insulin syringe with an insulin needle (0.3 mm outer diameter, BD Micro-Fine, Franklin Lakes, NJ, USA) was used to inject approximately 2 µL of 2.7% PAAH + PMs into the muscles of *D. rerio* in the lateral line region under the dorsal fin (n = 12) (Figure 1F). After that, the fish were immediately returned to aerated aquaria for recovery.

### 2.5. Histological Examination of Fish with Implanted pH-Sensitive Hydrogel

To study the immune response in dynamics, three individuals were randomly taken 6 h, 24 h, 7 days, and 14 days after gel injection and sacrificed by placing them in 0.05 mL/L solution of clove oil until the fish stopped reacting to the external stimulus. The tissues at the injection site were excised with a blade (Gillette, Boston, MA, USA) and fixed in Bouin’s solution (HT101128, Sigma-Aldrich, St. Loius, MO, USA) for a day, followed by washing in 70% alcohol until the yellow color of the tissue was eliminated. Samples were dehydrated with isopropanol (BioVitrum, Moscow, Russia) and impregnated with homogenized paraffin medium for histological studies (BioVitrum, Moscow, Russia) according to the scheme for endoscopic biopsy specimens [32]. The obtained paraffin blocks were sliced on a sledge microtome (MS-2, Moscow, Russia) with a cut thickness of 7 µm.

Before staining, histological sections were examined under a fluorescent microscope in the red channel to localize the site of injection by the fluorescence of microcapsules. Slides with tissue samples were then deparaffinized in xylene, placed in Mayer’s hematoxylin solution (BioVitrum, #05-022) for 4 min, transferred to tap water for 5 min, incubated in 0.5% water eosin (BioVitrum, #05-010) for 4 min, and washed in 96% ethanol and in xylene for 10 min before being embedded in Vitro gel epoxy resin (BioVitrum, Moscow, Russia). Histological sections were examined and visualized using a Mikmed-2 fluorescence microscope (LOMO, Saint Petersburg, Russia) with a 1200D camera (Canon, Taichung, Taiwan). Representative photos were automatically stacked using the Zerene Stacker software (v.1.04, Zerene Systems LLC, Richland, WA, USA), if necessary. The programs GIMP (v.2.10.20, The GIMP Team) and Inkscape (v.1.0.2, Inkscape’s Contributors) were used for image correction.

### 2.6. pH Measurements Using SNARF-1

The pH level in PAAH + PMs was monitored under Mikmed-2 fluorescence microscope (LOMO, Saint Petersburg, Russia) directly inside the syringe (during washing) or inside zebrafish tissues by recording the fluorescence spectrum of the SNARF-1 dye (Figure 1E). For this, a QE Pro spectrometer (Ocean Optics, Orlando, FL, USA; INTSMA-200 optical slit; accumulation time 1 s) was connected to the camera port of the fluorescence microscope using optical fiber, QP400-2-VIS-NI (Ocean Optics, Orlando, FL, USA), and an F280SMA-A collimator (Thorlabs, Newton, NJ, USA) [30]. The measurements were performed at the excitation wavelength of 545 nm, and the spectral signal was obtained in the red channel. The ratio of fluorescence intensities at two wavelengths of the SNARF-1 spectral peaks (605 and 640 nm) was used to measure pH after building the calibration curve using linear regression [33]. To construct the calibration curve, approximately 3 µL of PAAH + PMs were placed on a glass slide with approximately 10 µL of sodium phosphate buffers for at least 5 min to equilibrate pH inside the gel with the buffer. The fluorescence ratios I605/I640 were calibrated in the pH range of 4.5–8.2.

Besides the injection of 2.7% PAAH + PMs into the muscles of the zebrafish, each in vivo pH measurement under the microscope also required anesthesia with clove oil. Fluorescence spectra of SNARF-1 were acquired 3 h, 6 h, 1 day, 2 days, and 7 days after injection for measurements of interstitial pH at normal conditions. Before calculating I605/I640 ratios for implanted PAAH + PMs, background spectra of undamaged tissues (formed by light scattering and weak autofluorescence) were subtracted from the spectra of SNARF-1 using Scilab (v.5.5.2, ESI Group, Rungis, France).

### 2.7. Hypercapnic Exposure

In order to induce metabolic acidosis in muscle tissues, fish were exposed to water hypercapnia. The CO_2_ level in the water was determined using the WaterTest Set (Tetra, Melle, Germany) and the NILPA test (Saint Petersburg, Russia). After acquiring the SNARF-1 spectra 2 days post injection of PAAH + PMs, the fish were transferred not to tanks with normal water hydrochemistry (2 mg/L CO_2_) but into tanks (5 L) with CO_2_ saturated water (97 mg/L; n = 5). After one hour of exposure to hypercapnia, fish muscle pH was again measured as described above.

## 3. Results

### 3.1. In Vitro Characterization of 2.7% PAAH

In our study, a low-density semi-liquid polyacrylamide hydrogel was tested for injection into tight fish muscles. At the monomer concentration of approximately 2.5–3.5%, PAAH after polymerization forms a jelly-like viscous sticky substance that can be drawn or dispensed from a syringe many times. As shown by microscopic examination, 2.7% is close to the maximum monomer concentration at which the gel retained its semi-liquid microstructure; when pressed, the 2.7% PAA hydrogel formed elastic cloud-like compactions with rounded edges (Figure 1B). In contrast, the more concentrated 3.3% PAAH under pressure disintegrates into chunks with ragged edges (Figure 1C). Kinematic viscosity of 2.7% PAAH was found to be approximately 167 times higher than for glycerol, i.e., approximately 0.193 m^2^/s. It was also found that, despite its fluidity, when pressed and smeared, 2.7% PAAH retains incorporated fluorescent microcapsules, preventing them from protruding from the gel (Figure 1D). Thus, 2.7% PAAH was chosen for further experiments as it maintains its integrity under the mechanical force and protects PMs from contact with the external environment.

### 3.2. Histological Examination of the Immune Response to the Implanted Hydrogel in Fish Muscles

Analysis of tissue microstructure on the first day after injection mainly revealed damage to fish muscles caused by the injection procedure. In *D. rerio*, intact muscle fibers (myofibrils) have a regular fusiform shape and are tightly adjacent to each other. Six hours after the implantation procedure, the myofibrils at the site of injection were shortened, twisted, and fragmented due to mechanical damage (Figure 2). Around the gel, there was an accumulation of purple basophilic cells, almost entirely erythrocytes, indicating the rupture of microcapillaries at the site of injection. PAAH + PMs on histological sections was represented by a basophilic fibrous substance with inclusions of fluorescent dots visible in the red channel (Figure 2 and Figure 3). In the lumen of the injection channel, the hydrogel on the sections often gathered into folds and left empty spaces at the sites of attachment to the surrounding tissues. This inhomogeneity could be due to the dehydration during the histological processing.

The same picture of mechanical tissue injury was observed twenty-four hours after the injection of the gel. Around and within amorphous PAAH + PMs, no recruitment of lymphocytes to remove dead tissue was found, indicating that the inflammatory phase of wound healing has not yet begun (Figure 3). This is in general consistent with the literature data on the development of the hemostatic process in the first hours after injury during normal wound healing in fish [34,35].

One week after the treatment, no damaged myofibrils and blood-forming elements were found at the site of injection (Figure 4A), indicating the completion of the normal process of eliminating the consequences of mechanical tissue damage. However, around the implanted PAAH + PMs there was a pronounced inflammation characterized by the involvement of a large number of monocytes, eosinophils, and lymphocytes, as well as fibroblasts (Figure 4D). These immune cells remove necrotic tissue, phagocytize foreign bodies, and synthesize immune response mediators and nonspecific immune factors [36]. Furthermore, seven days after the injection, immune cells massively penetrated inside the PAAH + PMs and were engulfing microcapsules. The overall integrity of the hydrogel was violated: both small fragments and large pieces of gel surrounded by monocytes (granulomas) (Figure 4C,D) were observed. The described structures indicate partial degradation of the polymer and an ongoing cellular immune response to the introduction of the foreign body.

The same picture of chronic inflammation in the presence of a foreign object was observed in fish muscles at the end of the second week after the injection of PAAH + PMs (Figure 5). Loose tissue in the lumen of the injection channel mainly consisted of phagocytes with clusters of forming granulomas around the remnants of PAAH + PMs in the middle. In some places, phagocytes mixed with fibroblasts formed regularly arranged structures, which may be the precursors of granulation tissue that is normally formed at the final stages of wound closure. However, no other signs of the late stages of the foreign body elimination reactions, including the fibrous capsule formation around the implant [35], were observed 14 days after implantation (Figure 5). Therefore, according to the obtained results, the inflammation caused by PAAH + PMs continues until the complete decomposition of the gel, while chronic inflammation provoked by biocompatible materials in fish usually ends by the second week [35].

### 3.3. In Vivo Tests of the pH-Sensitive Hydrogel

In order to test the functional properties of PAAH + PMs as the structural base for the pH sensor, the SNARF-1 fluorescence was first monitored during 2 days post injection (Figure 6A). Previously, a relatively acidic pH of approximately 6.9 was observed directly after injection of free microcapsules with SNARF-1, along with an increase in the blood pH level over approximately 3 h [13], so here measurements were started after 3 h. When fish were kept under normal conditions (2 mg/L of CO_2_ in water), spectra of SNARF-1 within the implants corresponded to the median pH values of approximately 7.4–7.6, which is in agreement with the previous measurements for zebrafish blood [13]. One-way ANOVA did not reveal statistically significant deviations in the time range from 3 h to 2 days after injection (*p* = 0.223). Next, after the 48-h pH measurement, the animals were subjected to hypercapnic conditions for 1 h (97 mg/L CO_2_ in water). The readout of SNARF-1 inside implanted PAAH + PMs indicated a decrease in the median pH level from 7.44 at 48 h to 7.18 after the additional hour at elevated CO_2_ (Figure 6). The difference between these two measurements was statistically significant (*p* = 0.014). Thus, the obtained data demonstrate the sensitivity of this easily implantable sensor to changes in the interstitial pH for at least 2 days.

The histological analysis demonstrated at least partial engulfment of PMs within amorphous PAAH by the immune cells on the 7th day post injection. Because the conditions inside the lysosomes of phagocytes should be acidic, an additional experiment was performed in order to check the pH inside PMs at this time point. Unfortunately, it was not possible to perform these measurements due to an unexpected increase in autofluorescence at the site of injection. In the used optical setup, SNARF-1 at nearly physiological pH has the main peak of fluorescence at approximately 640 nm (Figure 6B), but after 7 days at the injection site, additional spectral peaks of fluorescence were found at approximately 675 and 740 nm. The analyzed individuals (n = 12) fell into the following three groups: (1) showing visibly normal spectra of SNARF-1; (2) showing spectra of SNARF-1 clearly mixed with relatively weak autofluorescence peaking at approximately 675 nm; (3) showing highly variable spectra with prevailing autofluorescence with peaks at 675 and 740 nm. Example spectra from the last group of individuals are presented in Figure 6B. The peaks at 675 and 740 nm could be both mixed within the same spectrum or presented separately. Thus, the observed autofluorescence was formed by at least two independent fluorophores. The peak at 675 nm may be due to lipofuscin, which can be accumulated in immune cells [37]. The fluorophore giving the peak at 740 nm may also be related to the native autofluorescence of immune cells, but the possibility that the cells are able to somehow chemically modify SNARF-1 in the engulfed PMs and the peak is due to this derivative fluorophore cannot be excluded.

Although we cannot fully explain the mechanism behind this autofluorescence, it is probably caused by the triggering of massive inflammatory processes around PAAH + PMs. Because at least a peak at 675 nm was found for most individuals and an independent spectrum of the respective fluorophore is not available, it made pH measurements with SNARF-1 unreliable in these conditions.

## 4. Discussion

In the course of our work, we investigated the in vivo durability of 2.7% polyacrylamide hydrogel used as a scaffold for polyelectrolyte microcapsules, which is of interest for the development of microsensors applicable to the body of small-sized fishes and, potentially, other animals. The choice of the gel formulation was due to the need to make a simple and cost-effective semi-liquid scaffold for microencapsulated fluorescent molecular probes to test it as a prototype for further microsensor research. Various non-resorbable but resilient hydrogels based on such polymers as poly(2-hydroxyethyl methacrylate) and poly(ethylene glycol) diacrylate, etc., with promising biocompatibility were previously applied as the matrix for implantable sensors and showed high performance [14,15,20,38,39]. Acrylamide has also been suggested as a useful component for tissue-integrating hydrogel implants [40,41]. Studies of resilient polyacrylamide-based gels (8% acrylamide concentration) have revealed low immune rejection of these materials in mammalian tissues, which can thus be used as a solid scaffold for cartilage repair. However, it was shown that the addition of other monomers (polyethylene glycol and methacrylated hyaluronic acid) can further increase the biocompatibility of these hydrogels [17,42].

Because aquatic animals cannot tolerate prolonged exposure to air, the microsurgery required to implant resilient hydrogels becomes particularly challenging for them, and injectable substances are more preferable, especially for small animals with tight tissues. In addition, according to earlier works, increasing water content and pore sizes in a hydrogel provides greater biocompatibility [43]. Thus, 2.7% semi-liquid polyacrylamide gel, due to its easy preparation procedure, was a reasonable choice as the implant material for initial tests in fish tissues.

Polyacrylamide gel with 2.5% monomer concentration was applied as a cosmetic filler in the 1990s; therefore, there are a number of studies of its tissue biocompatibility and long-term effects [44,45,46,47]. Studies of these hydrogels were mainly focused on cases of medical complications (mainly chronic inflammation and fibrosis), and their results were often controversial, with little discussion of the short-term consequences of the administration of 2.5% PAAH [48]. However, the time between PAAH injection and health complaints in humans was reported to be from six months to decades [44,46,47]. Such a period is mostly sufficient for application of implantable optical sensors in physiological research on fish and even for some potential tasks in aquaculture.

The histological approach was used in this study to evaluate the in vivo structural stability of 2.7% PAAH and to assess the course of inflammation caused by the implant. It was revealed that inflammatory processes associated with the penetration of immune cells inside the polyacrylamide hydrogel occurred in the period from the 1st to the 7th day after implantation. This is consistent with the development of significant autofluorescence at the site of injection, which was observed by the 7th day. The nature of this autofluorescence deserves further attention in the future, especially for the correction of its spectrum during measurements of any parameters using fluorescent molecular probes such as SNARF-1.

Examination of long-term biocompatibility demonstrates that semi-liquid PAAH is disintegrated by phagocytic cells and significantly slows down the inflammatory phase of the normal healing process [34,49]. Wound healing in adult fish is a complex multistage process including overlapping phases of hemostasis (thrombus formation), inflammation (lymphocyte recruitment and activation), proliferation (granulation tissue formation, re-epithelialization, and revascularization), and remodeling (collagen deposition and wound closure). Normally, the first phases last minutes and the second phase lasts up to 2 weeks, after which the repair process begins [34]. The presence of the foreign matter of PAAH + PMs caused chronic inflammation by the end of the second week after injection, indicating that the hydrogel was able to delay (in comparison to the free PMs) but not prevent recognition of the sensor by the immune system. It should be noted that it is difficult to assess the negative effect of delaying the immune system recognition and elimination of known hydrogels of different compositions, because the histological studies of the development of inflammation around the implants in dynamics are very scarce. However, the approach of combining PMs and PAAH as the double protection for molecular probes indeed was shown to be effective and useful because the immune cells need more time to reach PMs after the start of disintegration and penetration into the outer layer of PAAH.

Using 2.7% PAAH hydrogel implants, we were able to monitor the physiological pH values in *D. rerio* muscles for at least two days after injection. We also showed the sensitivity of the developed sensors to metabolic acidosis in fish tissues provoked by increased environmental CO_2_ during this period. Previously, we demonstrated that free polymeric microcapsules with polyethylene glycol coating are recognized by the immune system after hours and cause chronic inflammation for weeks [50]. Therefore, the obtained result demonstrates the increase in the operating time of implanted optical sensors based on polyacrylamide hydrogel up to days and confirms that amorphous PAAH delays the immune response of incorporated microparticles.

As the main result of this study, for the first time we were able to use hydrogel implants with fluorescent indicator dye to provide accurate mid-term monitoring of the critical physiological parameter inside the bodies of small fish. Previously, functional tissue-compatible implants were prepared for mammals [51] or were only suitable for large aquatic animals [15]. Importantly, the polyacrylamide carrier is easy to prepare and familiar to biologists worldwide, which is very important for introducing the proposed procedures into routines of physiological research. Thus, in this work, we have provided the methodological basis for the subsequent development of implantable devices with various functional applications based on amorphous hydrogel. Given the demonstrated convenience of using semi-liquid gels to inject, hold, and protect microencapsulated probes, the proposed approach can be subsequently applied using newly developed innovative synthetic semi-liquid hydrogel formulations with promising biocompatibility [28,52].

## Figures and Tables

**Figure 1 polymers-14-03956-f001:**
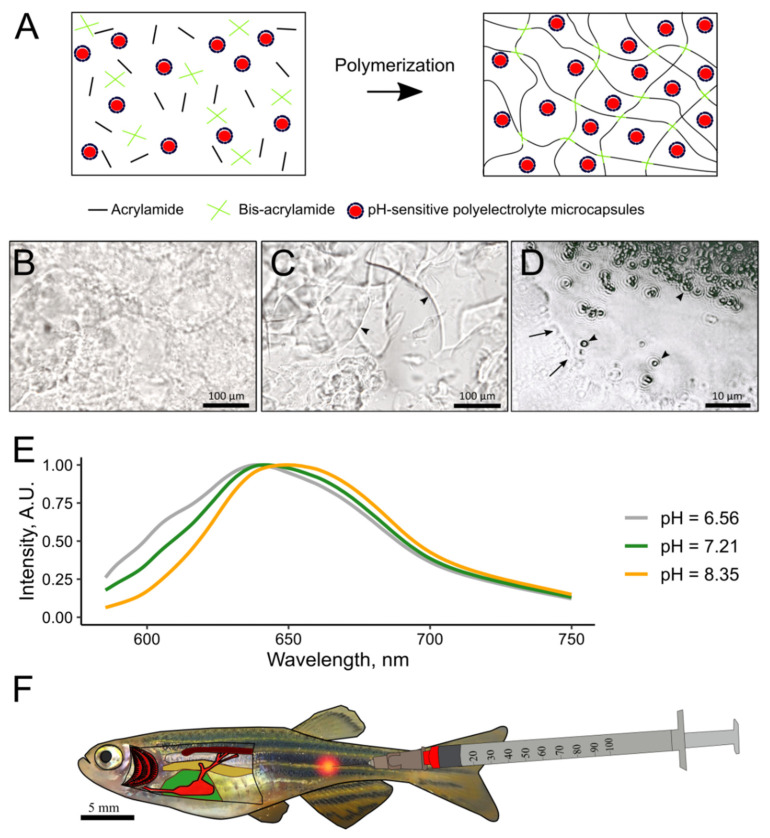
Preparation and use of PAAH + PMs with pH-sensitive SNARF-1. (**A**) a scheme of polymerization of PAAH with embedded PMs. (**B**) representative image of 2.7% PAAH crushed with a cover glass. (**C**) representative image of 3.3% PAAH crushed by a cover glass. Arrowheads indicate chips in the squeezed gel. (**D**) 2.7% PAAH with fluorescent PMs smeared on glass. Black arrows indicate the border of the gel. The arrowheads point to the microcapsules. (**E**) example fluorescence spectra of a PAAH + PMs with microencapsulated SNARF-1 at different pH, obtained using the red channel of a fluorescent microscope and normalized at peak intensity. (**F**) general plan of the fish body with the red dot marking the site of the PAAH + PMs injection.

**Figure 2 polymers-14-03956-f002:**
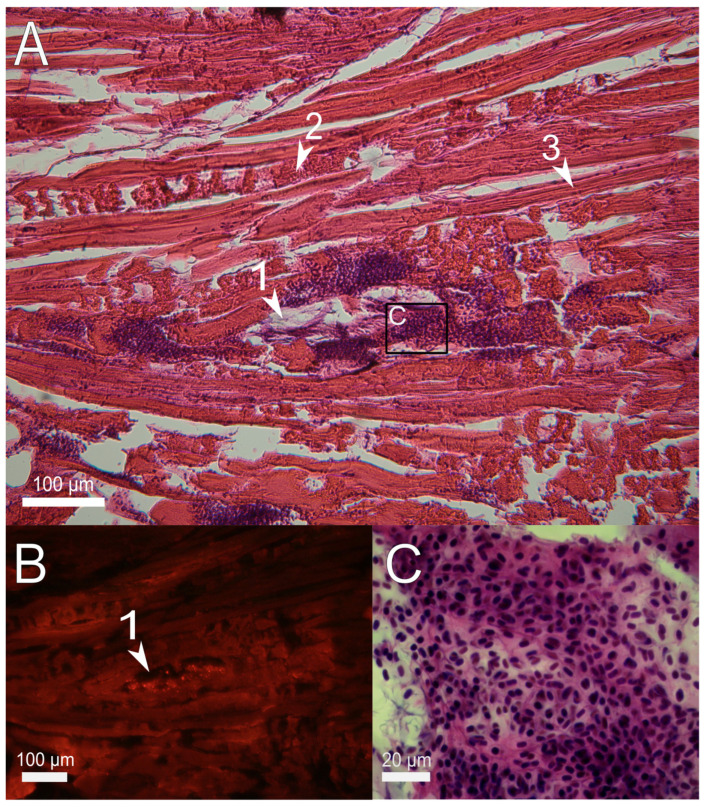
Sagittal histological section of *D. rerio* muscles 6 h after the injection of 2.7% PAAH + PMs. (**A**) histological section of the injection site (×10), H&E stain; (**B**) fluorescence of the unstained histological section in the red channel indicates microcapsules loaded with SNARF-1; (**C**) enlarged image of the region from A (×100) showing the aggregation of spilled erythrocytes (hematoma); 1, PAAH + PMs; 2, damaged myofibrils; 3, normal myofibrils.

**Figure 3 polymers-14-03956-f003:**
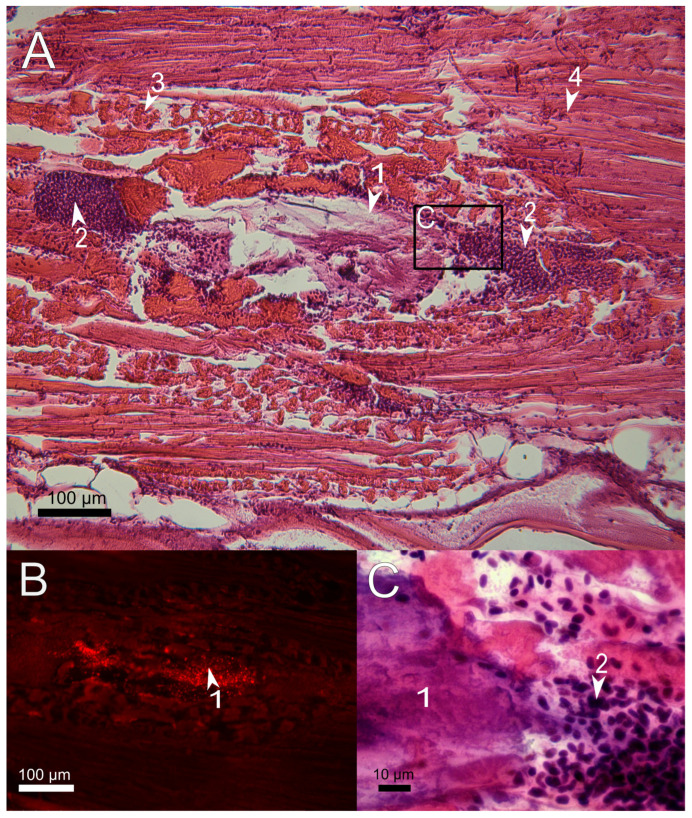
Sagittal histological section of *D. rerio* muscles 24 h after the injection of 2.7% PAAH + PMs. (**A**) histological section of the injection site (×10), H&E stain; (**B**) fluorescence of the unstained histological section in the red channel indicates microcapsules loaded with SNARF-1; (**C**) enlarged image of region from A depict the border between the hydrogel and the spilled erythrocytes (×100); 1, PAAH + PMs; 2, hematoma consisting of spilled erythrocytes; 3, damaged myofibrils; 4, normal myofibrils.

**Figure 4 polymers-14-03956-f004:**
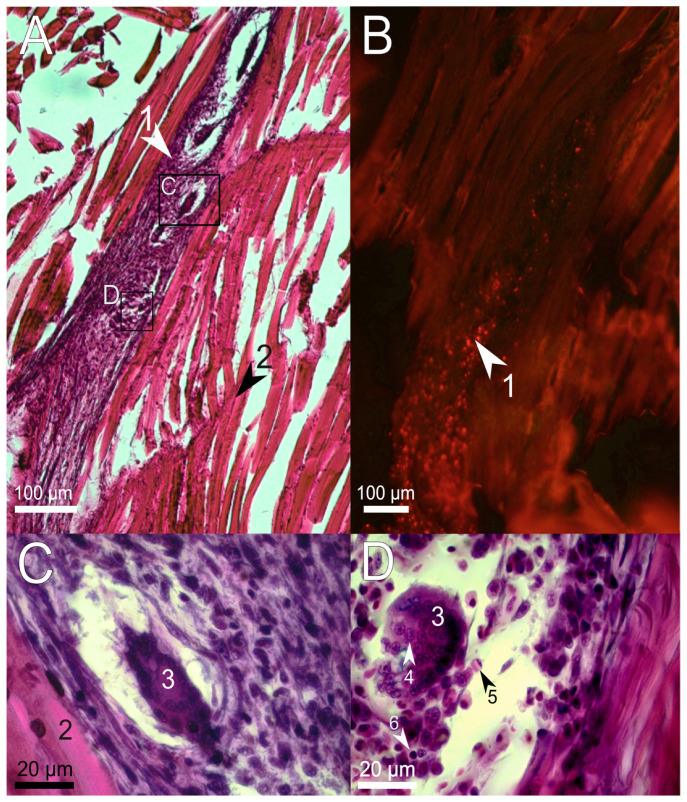
Sagittal histological section of *D. rerio* muscles 7 days after the injection of 2.7% PAAH + PMs. (**A**) histological section of the injection site (×10), H&E stain; (**B**) fluorescence of the unstained histological section in the red channel indicates microcapsules loaded with SNARF-1; (**C**,**D**) enlarged images of the region from A showing pieces of hydrogel surrounded by phagocytes; 1, PAAH + PMs; 2, intact muscle fibers; 3, granuloma formation; 4, monocyte; 5, eosinophil; 6, lymphocyte.

**Figure 5 polymers-14-03956-f005:**
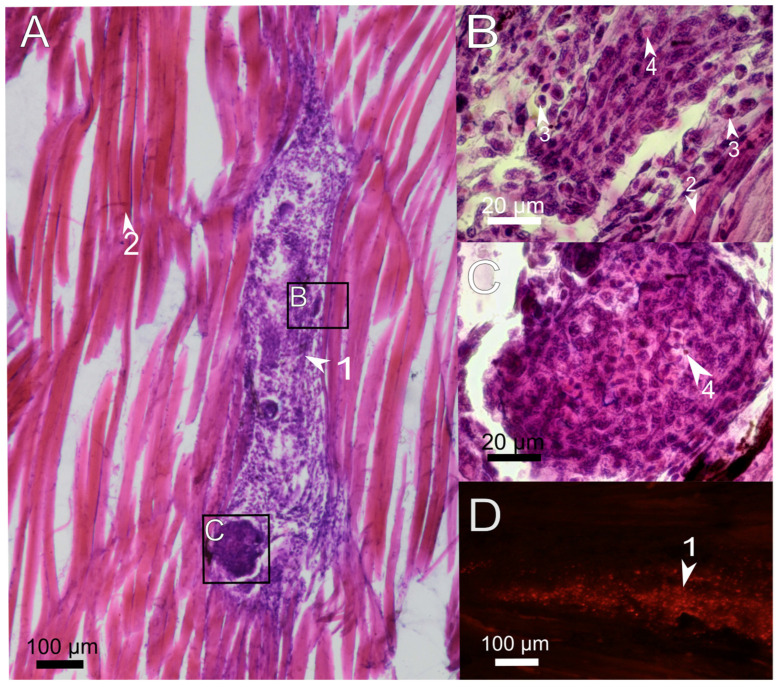
Sagittal histological section of *D. rerio* muscles 14 days after the injection of 2.7% PAAH + PMs. (**A**) histological section of the injection site (×10), H&E stain; (**B**,**C**) enlarged images of regions from A showing granulomas, which are dense agglomerations of phagocytic cells around remnants of PAAH. (**D**) fluorescence of the unstained histological section in the red channel indicates microcapsules loaded with SNARF-1. 1, PAAH + PMs; 2, muscle fibers; 3, eosinophilic granular cells; 4, fluorescent microcapsules.

**Figure 6 polymers-14-03956-f006:**
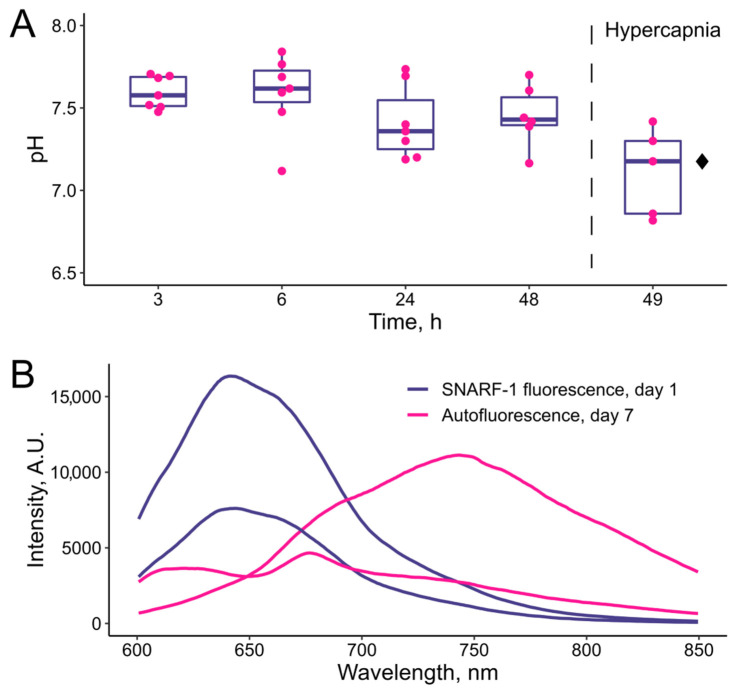
Testing long-term functionality of implanted PAAH + PMs with pH-sensitive SNARF-1. (**A**) pH monitoring in fish muscles using SNARF-1 spectra for two days at normal CO_2_ concentration in the water (2 mg/L) and after 1 h exposure to high CO_2_ (97 mg/L). Dots indicate individual measurements. Black diamond indicates statistically significant difference from 48 h according to the paired Student’s *t*-test with *p* < 0.05. (**B**) Example spectra of SNARF-1 immediately after implantation of PAAH + PMs and example spectra of autofluorescence (possibly combined with SNARF-1 fluorescence) at the injection site 7 days later. All spectra were obtained from the same individual.

## Data Availability

All the obtained data are available within the main text.

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
