# Peer review of "Durability of Implanted Low-Density Polyacrylamide Hydrogel Used as a Scaffold for Microencapsulated Molecular Probes inside Small Fish"

_polymers, 2022, doi:10.3390/polym14193956_

Round 1

Reviewer 1 Report

In the manuscript “Durability of implanted low-density polyacrylamide hydrogel used as a

scaffold for microencapsulated molecular probes inside small fish” by Maxim Timofeyev et al., authors introduced a hydrogel sensor based on amorphous 3% polyacrylamide hydrogel which could help trace the interstitial pH in the fish muscles under normal and  hypercapnic conditions. Overall, the article is comprehensive and well written. Some minor revisions should be addressed before the publication in this journal.

The raised concerns are listed below:

1. It is desirable to provide some schematic diagrams of hydrogel formation and functional execution

2. Authors should include some hydrogels whose mechanical properties were well controlled in the manuscript, such as “PNAS, 2021, 118 (37), e2110961118ï¼›Chinese Journal of Chemistry, 2021, 39, 2711-2717ï¼›Materials & design, 2020:108492”.

3. Authors should provide some test data to explain the mechanical characteristics of PAAH hydrogel in section3.1. For example, the rheological measurement.

4. In section3.3, what causes the peak at 740nm.

Author Response

Reviewer 1

Open Review
(x) I would not like to sign my review report
( ) I would like to sign my review report

English language and style
( ) Extensive editing of English language and style required
( ) Moderate English changes required
( ) English language and style are fine/minor spell check required
(x) I don't feel qualified to judge about the English language and style

Yes    Can be improved    Must be improved    Not applicable
Does the introduction provide sufficient background and include all relevant references?
(x)    ( )    ( )    ( )
Are all the cited references relevant to the research?
(x)    ( )    ( )    ( )
Is the research design appropriate?
(x)    ( )    ( )    ( )
Are the methods adequately described?
( )    (x)    ( )    ( )
Are the results clearly presented?
( )    (x)    ( )    ( )
Are the conclusions supported by the results?
(x)    ( )    ( )    ( )

Comments and Suggestions for Authors
In the manuscript “Durability of implanted low-density polyacrylamide hydrogel used as a scaffold for microencapsulated molecular probes inside small fish” by Maxim Timofeyev et al., authors introduced a hydrogel sensor based on amorphous 3% polyacrylamide hydrogel which could help trace the interstitial pH in the fish muscles under normal and  hypercapnic conditions. Overall, the article is comprehensive and well written. Some minor revisions should be addressed before the publication in this journal.

Authors: Dear Reviewer 1, we would like to thank you for critically reviewing the article and for the high estimate of our work. We hope that after the performed revision the manuscript became more understandable. Please check the attached file for more convenient reading.

The raised concerns are listed below:
1. It is desirable to provide some schematic diagrams of hydrogel formation and functional execution

Authors: We agree that more visual information about the used hydrogel will be helpful for readers, so now we significantly expanded Figure 1 by adding the scheme of PAAH polymerization and example spectra of microencapsulated SNARF-1 within the hydrogel. In particular, the fluorescence spectra of SNARF-1 inside PAAH+PMs at different pH should provide readers with a better explanation how the pH measurements (presented on Figure 6) were performed. Please have a look at the updated Figure 1 below (available in the attached file).

2. Authors should include some hydrogels whose mechanical properties were well controlled in the manuscript, such as “PNAS, 2021, 118 (37), e2110961118ï¼›Chinese Journal of Chemistry, 2021, 39, 2711-2717ï¼›Materials & design, 2020:108492”.

Authors: We significantly expanded our Introduction and Discussion sections now and included these and other new relevant references.

3. Authors should provide some test data to explain the mechanical characteristics of PAAH hydrogel in section 3.1. For example, the rheological measurement.

Authors: We would like to especially thank you for this suggestion since it helped us to present the information about the used hydrogel much better. First, we found a too strong rounding of the acrylamide concentration throughout the manuscript: 3 % instead of more exact 2.7 %. The correct and full information about the gel preparation was initially presented in section 2.2, but since the difference in viscosity between the gels with 3.0 and 2.7 % acrylamide concentrations is substantial, we corrected the value to 2.7 % in order to be more specific. Second, the rheological properties of the used gel are indeed very important for reproducibility of the results, and we tried to fulfill this suggestion. Unfortunately, we do not have any rheometers available right now, so we had to rely on the gravimetric method for measurement of kinematic viscosity. We found 2.7 % PAAH to have viscosity out of the dynamic ranges of the standard Ubbelohde viscometers we have available and used a vertical tube for the relative measurements in comparison to glycerol with known kinematic viscosity. Such a measurement is, of course, approximate (which we clearly acknowledge in the manuscript), but the obtained information should give the readers the necessary mechanical characterization of the used hydrogel. Please check the sections 2.2 and 3.1 for the added information.

4. In section3.3, what causes the peak at 740nm. 

Authors: We do not have a definite explanation for the autofluorescence at 740 nm, but we added to this section the following thoughts for what fluorophore it can be:
“The fluorophore giving the peak at 740 nm may also be related to native autofluorescence of immune cells, but we cannot exclude the possibility that the cells are able to somehow chemically modify SNARF-1 in the engulfed PMs and the peak is due to this derivative fluorophore.”

Submission Date
20 August 2022
Date of this review
29 Aug 2022 11:40:02

Thank you very much again for your time and the suggestions provided!

Very sincerely yours,
Prof. Maxim Timofeyev
Irkutsk State University

Reviewer 2 Report

This manuscript on the implantation of polyacrylamide hydrogels in fish seems like it could provide useful information to researchers in the field. The inclusion of histology data helps clarify some of the in-vivo behaviors of these hydrogel constructs. It is an interesting study that leaves open questions; this makes it read more like a communication.

It isn't entirely clear to the reviewer if the hydrogels are the novelty or if it is the use of encapsulated fluorescent dye in the fish? The introduction could be improved to clarify the place this work occupies in the field. More references to what other teams have done thus far would really help identify the importance of this work.

The discussion is a bit short and it seems like many of the conclusions are mostly observational. If this is not intended to be a communication a more thorough discussion of the results would help bring the manuscript to life.

Author Response

Reviewer 2

Open Review
( ) I would not like to sign my review report
(x) I would like to sign my review report

English language and style
( ) Extensive editing of English language and style required
( ) Moderate English changes required
(x) English language and style are fine/minor spell check required
( ) I don't feel qualified to judge about the English language and style

Yes    Can be improved    Must be improved    Not applicable
Does the introduction provide sufficient background and include all relevant references?
( )    (x)    ( )    ( )
Are all the cited references relevant to the research?
(x)    ( )    ( )    ( )
Is the research design appropriate?
(x)    ( )    ( )    ( )
Are the methods adequately described?
(x)    ( )    ( )    ( )
Are the results clearly presented?
(x)    ( )    ( )    ( )
Are the conclusions supported by the results?
( )    (x)    ( )    ( )

Comments and Suggestions for Authors
This manuscript on the implantation of polyacrylamide hydrogels in fish seems like it could provide useful information to researchers in the field. The inclusion of histology data helps clarify some of the in-vivo behaviors of these hydrogel constructs. It is an interesting study that leaves open questions; this makes it read more like a communication.

Authors: Dear Reviewer 2, we would like to thank you for critically reviewing the article and for the high estimate of our work. We hope that after the performed revision the manuscript became more complete. Please check the attached file for more convenient reading.

It isn't entirely clear to the reviewer if the hydrogels are the novelty or if it is the use of encapsulated fluorescent dye in the fish? The introduction could be improved to clarify the place this work occupies in the field. More references to what other teams have done thus far would really help identify the importance of this work.

Authors: We would like to especially thank you for this suggestion since we now noticed that Introduction indeed missed several important points. The novelty of our study is the combination of semi-liquid hydrogel and microcapsules as the combined structural base for sensing implants in tight animal tissues. Free PMs and especially shaped hydrogels (i.e. resilient, not semi-liquid) separately were of course previously applied in a number of studies for sensing purposes (despite only a couple of them included fish), but not amorphous hydrogels with embedded PMs. We now substantially expanded the Introduction section and added a number of relevant references in order to better place our work in the context of modern literature. We hope that Introduction now became more understandable.

The discussion is a bit short and it seems like many of the conclusions are mostly observational. If this is not intended to be a communication a more thorough discussion of the results would help bring the manuscript to life.

Authors: The “Instructions for Authors” of Polymers states that the manuscript type “Communication” can be considered, but the main type for original research manuscripts in the journal is “Article”. Thus, despite we indeed thought about “Communication” as an option, we tried to adhere to the type “Article”. In the revised version of the manuscript we significantly expanded the Discussion section and added new citations on histological studies of other hydrogels, including those based on polyacrylamide. We also tried to connect the motivation of our research in Introduction with certain conclusions in Discussion; for example, we now highlight that the approach of combining amorphous hydrogel with embedded microcapsules was indeed found useful. We hope the conclusions now look more supported.

Submission Date
20 August 2022
Date of this review
30 Aug 2022 21:59:46

Thank you very much again for your time and the suggestions provided!

Very sincerely yours,
Prof. Maxim Timofeyev
Irkutsk State University

Round 2

Reviewer 2 Report

looks like the authors have injected some life into this manuscript with a more thorough framing of the work and some new discussion points. the study design and approach in general is of high quality and the manuscript seems to better reflect the hard work and intellectual merit of the authors now.